# Challenges in Working Conditions and Well-Being of Early Childhood Teachers by Teaching Modality during the COVID-19 Pandemic

**DOI:** 10.3390/ijerph19084919

**Published:** 2022-04-18

**Authors:** Kyong-Ah Kwon, Timothy G. Ford, Jessica Tsotsoros, Ken Randall, Adrien Malek-Lasater, Sun Geun Kim

**Affiliations:** 1Jeanine Rainbolt College of Education, University of Oklahoma, Norman, OK 73019, USA; tgford@ou.edu (T.G.F.); sgkim1112@ou.edu (S.G.K.); 2College of Allied Health, University of Oklahoma Schusterman Center, Tulsa, OK 74135, USA; jessica-tsotsoros@ouhsc.edu (J.T.); ken-randall@ouhsc.edu (K.R.); 3Department of Teaching Learning and Curriculum, University of North Florida, Jacksonville, FL 32224, USA; a.malek@unf.edu

**Keywords:** COVID-19 impact, early care and education, early childhood teachers, well-being, job demands, teaching modality, racial disparity

## Abstract

While a global understanding of teacher well-being during the COVID-19 pandemic is beginning to emerge, much remains to be understood about what early childhood teachers have felt and experienced with respect to their work and well-being. The present mixed-method study examined early care and education (ECE) teachers’ working conditions and physical, psychological, and professional well-being during the COVID-19 pandemic using a national sample of 1434 ECE teachers in the U.S. We also explored differences in working conditions and well-being among in-person, online, and closed schools, given the unique challenges and risks that ECE teachers may have faced by teaching in these different modalities. From the results of an online survey, we found that in the early months of the pandemic, many ECE teachers faced stressful, challenging work environments. Some were teaching in new, foreign modes and formats, and those still teaching in person faced new challenges. We found many common issues and challenges related to psychological and physical well-being across the three teaching groups from the qualitative analysis, but a more complicated picture emerged from the quantitative analysis. After controlling for education and center type, we found that aspects of professional commitment were lower among those teachers teaching in person. Additionally, there were racial differences across several of our measures of well-being for teachers whose centers were closed. Upon closer examination of these findings via a moderation analysis with teacher modality, we found that Black and Hispanic teachers had higher levels of psychological well-being for some of our indicators when their centers were closed, yet these benefits were not present for Black and Hispanic teachers teaching in person.

## 1. Introduction

It has been well documented that the majority of early care and education (ECE) teachers reported high levels of satisfaction with and commitment to their work before COVID-19 [1,2,3]. However, other barriers and challenges—disparities in wages, benefits, resources, and challenging working conditions—outweigh satisfaction and commitment and serve as job stressors [2,4,5,6]. Unfortunately, the COVID-19 pandemic has likely exacerbated challenges to teachers’ work and well-being [7,8]. The pandemic necessitated rapid changes in teaching and student support—the demands of which fell squarely on the shoulders of teachers and leaders—and were intensified by a shifting landscape as schools and communities were pressed to re-open [9,10,11,12]. Such demands have resulted in unprecedented stress, threatening the short- and long-term well-being of teachers, many of whom are coping with similar stress and demands in their personal lives [9].

Papaioannou et al. [13] have referred to the COVID-19 period as a triple pandemic, calling attention to not only the disease itself, but to the physical inactivity and mental illness that have followed. Numerous studies conducted during the pandemic have slowly revealed the deleterious effects of social restrictions, “shelter-at-home”, and online learning on adults [14]. The general consensus of the empirical research on P-12 teachers conducted in various countries around the world, such as the U.K., the U.S., Brazil, Mexico, Australia, Spain, and Portugal, is that teachers have suffered a significant impact to their psychological, physical, and professional well-being [8,15,16,17,18]. For example, Swigonski et al. [8] found that physical and behavioral symptoms of stress among early childhood teachers in the U.S. were 2–3 times higher during the COVID-19 pandemic, and this is significantly higher than in the community at large. Similarly, Alves et al. [15] found that the pandemic has reduced teachers’ perceptions of professional well-being, leading them to feel more uncertain about the future of their career.

The role of teachers and ECE centers/schools becomes even more significant in crisis situations such as the COVID-19 pandemic as students and families look to teachers for more psychosocial support [19]. For example, Ozmiz-Etxebarria et al. [19] found that preschool and primary grade teachers working in a university nursery school showed the highest ratings of psychological symptoms such as anxiety. Teachers and ECE centers/schools are expected to serve as the “great equalizer”, providing additional social-emotional learning and educational opportunities for vulnerable students who are more likely to be harmed in such crises [20,21]. Thus, it is critical to broadly examine ECE teachers’ working conditions and well-being during the pandemic, but holistic studies of working conditions and well-being remain sparse [15], particularly for U.S. teachers and U.S. ECE teachers more specifically [11,22]. However, few studies with national samples are available.

The onset of the COVID-19 pandemic meant that many ECE centers and schools transitioned to online learning, some closed altogether, and others remained open, precipitating substantial upheaval in the nature and scope of teachers’ professional lives and work. While evidence shows that teachers’ well-being has suffered due to the pandemic (e.g., declining well-being and satisfaction) [15], some studies are have revealed that these impacts may vary by different contextual and individual factors, such as teaching modality and the demographic characteristics of teachers. During the early months of the pandemic, some teachers suddenly found themselves employed without work, while others experienced increased workloads [23,24]. Teachers who remained teaching in person during the pandemic were at higher risk for contracting COVID-19 and at higher risk of severe illness if they did because of the chronic health conditions that accompany this vulnerable workforce [24,25]. The unique challenges and circumstances that surrounded the in-person work of ECE teachers likely influenced these risks as well. For instance, having a group of children in a small space, or having children in close contact during routine activities (e.g., diapering, feeding), likely made social distancing a challenge [23,24,25]. For ECE teachers, the additional tasks of taking precautions and constantly reminding young children to keep separated, wear masks, and wash hands was undoubtedly exhausting and stressful. For those one-in-four teachers at risk for severe illness, not remaining protected could have had lasting consequences, yet many ECE teachers may have felt financially compelled to continue to teach in person, given the low wages that many teachers receive, despite the risks [25].

Outside of worrying about their own health risks, some evidence suggests that teachers teaching online may have faced a unique set of stressors. For example, a study by Besser and his colleagues [26] demonstrated that a sudden transition to online teaching was related to higher levels of psychological stress among teachers. An international sample of 600 language K-12 teachers involved in online teaching also revealed high levels of stress related to additional workload, family health, loss of control over work decisions, blurred professional and personal lines, concerns about their colleagues and most vulnerable students, social isolation, and the stress of online teaching itself [27]. Similarly, Allen et al. [28] found high levels of stress for K-12 teachers in the U.K. teaching online during the beginning of the pandemic; however, those teaching in person were shown to have higher levels of stress for longer periods of time. Although ECE teachers teaching online are likely to face similar stressors, there is limited evidence in how these conditions affected them.

Some teachers worked in schools that were completely closed for a time during the pandemic, although this group of teachers rarely received attention. Teachers reported worries related to financial and job security as well as the sustainability of their center during closure [8,17]. Other teachers were worried about their students: how they were managing the pandemic, how they were being taken care of, and how their social development was being impacted. These types of worries could be associated with higher levels of poor well-being such as stress and vicarious/secondary trauma [17].

The picture of teachers’ working conditions and well-being differing by modality may be further complicated by teacher demographic characteristics and other contextual factors. Based on the well-documented fact that COVID-19 has disproportionately impacted people who are vulnerable, including children and adults from minoritized and under-resourced groups [29,30,31], we expected that teachers from underrepresented (e.g., racial/ethnic minority groups) and under-funded groups (e.g., private childcare centers, family childcare centers) were more likely to suffer from poorer working conditions and well-being during the pandemic. However, most extant studies during this period address racial/ethnic health disparities or educational inequality among underrepresented groups of children or the public—not teachers (see Souto-Manning and Melvin [32] for an exception). Limited empirical evidence has been collected across states documenting the shared and divergent impacts of the COVID-19 pandemic on the ECE workforce by teaching modality and the complex interplay among individual and contextual factors on teachers’ work and various aspects of well-being. In particular, research on ECE teachers’ physical well-being during the pandemic is scarce.

As one of the few exceptions, Collie’s study examined these issues, but in primary and secondary education settings [9]. In this study, Australian teachers’ gender, work experiences, and teaching modality were associated with differences in teacher working conditions and well-being. For example, teaching half remotely and half in person was related to greater stress. While teachers who have reduced work hours may have more time to address challenges in their teaching, it may also precipitate other concerns (e.g., concern about financial issues and job security due to reduced hours [9]). Despite its contribution, this study was conducted in primary and secondary school settings in Australia, which may not capture the unique demographic characteristics of the early childhood workforce (e.g., no variation in gender; more racial/ethnic diversity; varied educational level; more limited resources) in various settings (e.g., family childcare homes, private childcare, Head Start, public school, private school). In the U.S., Souto–Manning and Melvin [32] conducted an in-depth multi-method study to address these gaps by examining racial, occupational, and environmental factors on physical and psychological well-being among early childhood teachers of color in New York. However, the scope was limited to ECE teachers of color in one city.

### The Present Study

While a global understanding of teacher well-being during the COVID-19 pandemic is beginning to emerge, much remains to be understood about what ECE teachers have experienced and felt with respect to their working conditions and well-being—particularly considering the wide range in responses of ECE centers/schools to the pandemic in the U.S. Most of the existing evidence is limited in scope (e.g., confined to one region or one particular aspect of well-being), rarely addressing issues specific to ECE teachers who were already vulnerable prior to the pandemic. Furthermore, given that there was substantial variation in how teachers were teaching during this time, it is highly probable that teachers teaching in different modalities experienced different challenges in their working conditions and well-being and that these experiences might have differed across teachers’ race, educational level, and program type.

Thus, in this study, we report on the challenges the COVID-19 pandemic via quantitative and qualitative analysis of national online survey data from 1434 ECE teachers in the U.S. We asked three research questions: (a) How did ECE teachers’ working conditions and well-being differ by teaching modality (i.e., in-person, online, closed school)? (b) Were there teacher demographic and center-based differences in teacher well-being? and (c) Did teaching modality moderate the relationship between demographic and center-based differences and teacher well-being? The findings of this study could offer important implications for the field given the ongoing and unpredictable nature of the pandemic and its lingering effects.

## 2. Method

### 2.1. Participants and Settings

A total of 1434 ECE teachers serving children ages 0 to 5 (including Kindergarten) in 46 states in the United States completed an online survey in late spring to mid-summer of 2020. The overall racial/ethnic composition of the sample is similar to the population of early childhood teachers nationally [33], with a somewhat higher percentage of Hispanic teachers. The sample includes 58% White, 21% Hispanic, 14% Black, 3% American Indian or Alaska Native, 2% biracial, and 1% Asian. The vast majority of teachers in the sample were women (98.3%). The average age of the participants was 42 (*Range =* 17 to 80). The majority of teachers in the sample were fully paid (83%), but some were only partially paid (12%) or not paid at all (5%). Among those who were paid, the annual salary for fifty-one percent of the teachers was below USD 30,000. Fifteen percent of teachers received some form of public support, such as Medicaid, food stamps, or childcare subsidies. Seventy-three percent of participating teachers held an associate degree or higher, followed by some college but no degree (20%), and high school diploma or general education diploma (GED) (5%).

Participating teachers worked in Head Start programs (43%), childcare centers (34%), public schools (14%), family childcare homes (6%), and private schools (3%). They served infants and toddlers (24%), preschool or pre-K (38%), Kindergarten (6%), and children in multiage groups (31%). Teachers in the sample served children and families of diverse socio-economic status (SES): predominately low SES (54%), middle SES (15%), upper SES (7%), and mixed SES classrooms (24%). Of the 1434 early childhood teachers in the sample, approximately 29% reported that they were teaching in person, 28% were teaching online, and the remaining 43% were not teaching due to their centers/schools being closed.

### 2.2. Research Procedure and Analysis

After receiving Institutional Review Board (IRB) approval, we recruited early childhood teacher participants via various social media platforms (e.g., Facebook, Twitter). To ensure responses from various states and types of settings, such as private childcare centers, public schools, Head Start programs, and family childcare homes, stratified sampling (by state and setting type) was also integrated into the recruitment approach. This procedure involved producing a sample frame of ECE settings in each U.S. state and, from this, developing a contact list, which was first proportional to state population and then sought to preserve U.S. representativeness by setting type. These centers/schools were then first contacted to participate in the study. Once this procedure was complete, we also emailed state and national ECE organizations and agencies to distribute our survey more broadly.

Our interdisciplinary research team developed questions for an online survey that asked about their personal and professional background and the teachers’ experiences and well-being at work during the COVID-19 pandemic. The questions included: (a) demographic and background information, including teacher race, education, income; (b) teaching modality (i.e., teaching online, teaching in person, school closed); (c) whether they experienced changes in their work and well-being (i.e., negative change, positive change, no change); (d) if they experienced any change, what change they experienced (open-ended response); and (e) what they needed for improvement in their work (open-ended responses) and well-being (multiple choice). The online survey also consisted of previously validated scales to assess teachers’ well-being. On average, it took 25–30 min to complete the online survey. Among teachers who completed the survey and who requested to participate, sixty teachers were randomly selected to receive a USD 50 electronic gift card.

### 2.3. Measures

Below, we briefly describe the key measures used for the quantitative analysis of the study. More detailed descriptions and the psychometric properties of each measure are organized in Table 1.

#### 2.3.1. Working Conditions

This was assessed using three subscales from the Job Content Questionnaire (JCQ) [34]. The three subscales used in this study were physical job demands (related to the physical demands of the job), skill discretion (related to the variety of skills used for the job), and decision authority (related to the amount of job control). We also included questions asking teachers to report how they were paid (i.e., fully paid, partially paid, not paid) and if they had health insurance provided by their employer during the pandemic.

#### 2.3.2. Teacher Well-Being

**Psychological Well-Being**. Teachers’ psychological well-being was operationalized via ECE teachers’ perceptions of: (a) depressive symptoms, (b) stress, (c) resiliency, (d) life satisfaction, and (e) secondary trauma. Teacher depressive symptoms were assessed with the 10-item Center for Epidemiologic Studies of Depression Short Form (CES-D-10) [35]. Perceived stress was measured using the Perceived Stress Scale (PSS) [36]. Teachers’ resiliency was measured using the Brief Resilience Scale [37], and life satisfaction was assessed on the Satisfaction with Life Scale [38]. Lastly, teachers’ secondary trauma was assessed using one of the subscales of the Professional Quality of Life Scale [39] We used a total score of each measure for data analysis.

**Physical Well-Being**. Teachers’ physical well-being constituted a measure of (a) general health condition, (b) ergonomic pain, (c) food security, (d) Body Mass Index (BMI) [40] and (e) physical activity. To examine their general health status, we used a composite score of various dichotomous doctor-diagnosed symptoms that teachers reported. Ergonomic pain was assessed using the modified version of the Work-Related Musculoskeletal Disorders Scale (WMDS) [41]. The Short Form of the Food Security Survey Module was modified to assess food insecurity. Body Mass Index (BMI) [40] was calculated by dividing weight in kilograms by the square of height in meters. To measure physical activity, direct questions about how many days and hours were spent on vigorous physical activities and how much time was spent sitting on a weekday were asked.

**Professional Well-Being**. Professional well-being was assessed via two constructs: work commitment and intent to leave. Work commitment was measured using the Early Childhood Job Satisfaction Survey (ECJSS) [42]. Intent to leave the field/profession was measured via three items.

**Table 1 ijerph-19-04919-t001:** Description of measures used in the study.

Construct	Study Variables	Instruments	Instrument Characteristics	Psychometric Properties
Job Demands	Job demands	The physical demand, skill discretion, and decision authority subscale of the Job Content Questionnaire (JCQ) [34]	JCQ consists of 11 items in three domains of stress: demands (4 items), skill discretion (related to skill variety) (3 items), and decision authority (related to job control) (4 items). Subscales measure participants’ rating on how each statement was true of their work in ECE settings based on a 5-point scale: 1 (strongly disagree) to 5 (strongly agree).	The internal consistency for the scales of the original JCQ range Cronbach’s α = 0.65 to 0.79 [43]. Our study: JCQ physical demands subscale Cronbach’s α = 0.71.
Job Resources	Instrumental resources (wages and health insurance)	Questionnaire	One question asking teachers to report how they were paid (i.e., fully paid, partially paid, not paid) and one question asking teachers if they had health insurance covered by their employer during the pandemic (yes/no).	
Profess. Well-Being	Work commitment	Questionnaire from Early Childhood Job Satisfaction Survey (ECJSS) on work commitment [42]	ECJSS questions the range of personal and organizational factors related to employee satisfaction and work commitment in center-based ECE programs. Scores could range from 0 (low) to 10 (high levels of work commitment).	Overall consistency: Cronbach’s α = 0.89 [44]. Internal consistency for the commitment section: Cronbach’s α = 0.80 [45]. Our study: α = 0.74.
	Intent to leave	Questionnaire	Three questions that asked participants to rate their intent to leave the field on a 5-point scale (1 (very unlikely) to 5 (very likely)).	Our study: Factor loadings = 0.84 for all three items. Cronbach’s α = 0.82.
Psych. Well-Being	Depressive symptoms	Center for Epidemiologic Studies of Depression Short Form (CES-D-10) [35]—shortened version [46]	10-item screening test asking respondents to reflect on the previous week and rate the frequency of symptoms on a scale of 0 (not at all or less than 1 day) to 3 (5–7 days).	CES-D-10; Cronbach’s α = 0.65–0.91; [47,48]. Our study: Cronbach’s α = 0.75.
	Personal stress	The Perceived Personal Stress Scale (PSS) [36]	PSS items include questions about stress and examine how unpredictable, uncontrollable, and overloaded respondents find their lives. Other items ask participants about feelings during the last month, and the frequency of their feelings on a 5-point scale (1 = rarely/never to 5 = very often).	PSS; Cronbach’s α = 0.84; test-retest reliability Pearson *r* = 0.85 [36]. Our study: Cronbach’s α = 0.85.
	Life satisfaction	Satisfaction with Life Scale [38]	Five items measuring global cognitive judgments of a person’s life satisfaction on a 7-point scale based on how much they agree (1 = strongly disagree to 7 = strongly agree).	Test–retest reliability: Cronbach’s α = 0.82, coefficient Cronbach’s α = 0.87 [38] Internal consistency Cronbach’s α = 0.74 [49].
	Secondary trauma	A Subscale of Professional Quality of Life Scale (PROQOL) [39]	Thirty items organized in 3 subscales (compassion satisfaction, burnout, and secondary traumatic stress) that measure the negative and positive effects of helping others who experience suffering and trauma. Our study used the subscale of secondary traumatic stress.	Secondary traumatic stress scale Cronbach’s α = 0.81 [39].
	Brief Resiliency	Brief Resilience Scale [37]	Six items that assess one’s ability to bounce back or recover from stress measured on a 5-point scale (1 (strongly disagree) to 5 (strongly agree)).	Internal consistency Cronbach’s α = 0.80–0.91; test–retest reliability *r* = 0.62 [37].
Physical Well-Being	General health risk	General Health Indicator Composite	Eleven binary items (no = 0 and yes = 1) Items ask about doctor-diagnosed symptoms (e.g., anxiety, depression, infectious disease, heart disease). Our study used a total score of health conditions by combining all items.	
	Ergonomic Pain	Modified Work-Related Musculoskeletal Disorders Scale (WMDS) [41]	Five binary items (no = 0 and yes = 1) that ask about experienced pain in neck, back, shoulder, knee, and other. Our study used a total score by combining all items.	WMDS; Cronbach’s α (entire questionnaire) = 0.90; test–retest reliability Pearson *r* > 0.75 [41]. Our study: Cronbach’s α = 0.80.
	Food Security	Modified Food Security Measure (USDA) [26]	Five items that identify food-insecure households and households with very low food security with reasonably high specificity and sensitivity. Participants can score as high or marginally good security (0–1), low food security (2–3), very low food security (4–5). Two categories of “low food security” and “very low food security” mean food insecurity.	Reliability *α* = 0.75, Cronbach’s α = 0.91 [50].
	Obesity	Body Mass Index (CDC) [40]	Body Mass Index (BMI) is a person’s weight in kilograms divided by the square of height in meters.	
	Physical Activity	Questionnaire	Questions ask how many days and hours are spent on vigorous physical activities and how much time is spent sitting on a weekday.	

### 2.4. Data Analysis

We conducted an analysis of both qualitative and quantitative responses. For the analysis of qualitative data (i.e., open-ended responses in the survey), five members of the research team conducted a content analysis of the qualitative data (i.e., teachers’ open-ended responses in the online survey). The first author served as master coder and assigned questions to four other researchers for analysis. She met with individual coders, conducted open coding, and developed the initial codes. We (the first author and four other coders) met multiple times to refine codes and categories and discuss any discrepancy until reaching consensus. We conducted reliability checks with 10–15% of cases per question and established an inter-coder reliability ranging from 90 to 100 percent agreement before independent coding. Cohen’s Kappa ranged from 0.65 to 0.85 across the categories and coders. We compared categories of challenges in and needs for job demands and well-being during the pandemic across the three teaching modalities. While the response consolidation process resulted in more than ten categories, in this paper, we present the five most frequently reported responses from the content analysis. Similarly, the analysis of the quantitative data began with a descriptive analysis, and sample descriptive statistics are displayed in Table 2. Finally, an Ordinary Least Squares (OLS) regression analysis of the main effects of teachers’ education level, race, and type of setting on the various studied aspects of physical, psychological, and professional well-being was conducted. This was followed up by a test of the moderation of teaching modality upon the effects of teacher race and well-being.

## 3. Results

Of the teachers surveyed, approximately 29% reported teaching in person, 28% teaching online, while the remaining 43% were not working due to their centers being closed. As data were collected in the early phase of the COVID-19 pandemic, most of the teachers fit in one of these categories. Teaching modality significantly differed by type of setting: the majority of teachers in family childcare homes (82%) and in childcare centers/preschools (55%) taught in person, while 81% of public school teachers taught online. Fifty-four percent of Head Start teachers reported that their centers were closed during the pandemic and 39% of Head Start teachers taught online.

We found that in the early months of the pandemic, while many ECE teachers were committed to their work, moderately resilient, and tried to have a positive outlook, they still faced stressful, challenging work environments. Some were teaching in new, foreign modes and formats, and those still teaching in person faced new challenges. An understanding of these unique challenges begs an understanding of how they might have affected their overall well-being. Below, we examine this question with a specific focus on the status of ECE teachers’ job demands, as well as their physical, psychological, and professional well-being and how it differed by teaching modality.

### 3.1. Working Conditions and Needs by Teaching Modality during the Pandemic

From the content analysis (see Table 3), we found that, regardless of teaching modality, early childhood teachers experienced significant challenges working with young children in the early days of the pandemic, characterized by limited resources and a lack of clear guidelines and regulations. Although some challenges were common, we found distinctive challenges faced between the in-person teaching group and the online teaching group. As the school closed group was not working during the pandemic, our analysis focused only on the in-person and online teaching groups.

Respondents teaching in person reported many challenges at work: (a) challenges teaching and interacting with children and families in person during the pandemic; (b) financial hardships; (c) fears and uncertainty of becoming infected and passing it to their family; (d) additional job demands, and (e) too frequent changes in regulations and circumstances.

First, the major issue that many teachers encountered was related to the various challenges that arose in supporting and properly working in person with children and families during the pandemic while having to wear masks, implement social distancing, and frequently wash their hands. They also reported concerns about the current method of teaching not being optimal and developmentally appropriate for young children. It was difficult for teachers to help young children understand the situation and the importance of keeping masks on. One teacher stated:

*I work with infants. Wearing a mask makes it difficult to show them facial expressions. Infants love to see people smile at them… The infants don’t know the new teachers and don’t feel as safe as they would with their original teacher*.

In addition, teachers were concerned that their interactions and communications with families were limited or restricted because of health concerns during the pandemic. At the time, many centers did not allow families to enter the building so teachers were not able to have any face-to-face communication with families.

The second most frequently mentioned challenge was financial in nature. Low enrollment seemed to be a major cause of the financial issues of the center/school. While lower enrollment led to some positive changes for teachers (such as reduced workload and more time and attention given to each child), it also led to reduced working hours and compensation, which raised concerns for teachers. Furthermore, centers/schools were required to purchase additional resources such as cleaning supplies and personal protective equipment, which were limited in availability. This resulted in additional financial burdens and stress on the center as well as teachers and was particularly the case in family childcare homes and private childcare centers that relied heavily on tuition as a main source of income. One teacher commented, “*The biggest challenges of teaching during the COVID-19 is not having enough kids to stay open, but being shut down for a while, because we had no kids*”.

Third, although they tried to take all possible precautions, teachers who worked in person faced the great risk of becoming infected and passing it to their family. They reported that they were fearful and concerned about the possibility of contracting COVID-19 at work. Teachers mentioned that although they were aware of the lower risk of children being infected with COVID-19, they were afraid that a child could be infected without symptoms and inadvertently infect them. In addition, they were concerned that the supplies and resources provided were not often appropriate or sufficient to protect them from infection. One teacher said:

*I have to go through and to breathe and communicate clearly with the face mask on. The gloves I feel are acceptable but the mask that the company provided is not appropriate. It is a t-shirt that someone made., It is super thin almost see-through if you put up to the light. When I have it on and breathe the clothing goes into my mouth. Not sure if it is protective enough*.

Fourth, additional job demands were identified as a major challenge. These mostly consisted of additional tasks related to the new health and safety measures for in-person teaching, such as difficulty finding appropriate cleaning, personal protective equipment, and materials (as they were often out of stock or in low supply); having to follow many new regulations regarding cleaning/safety, and having to do constant cleaning. Pick-ups and drop-offs to ensure safety became challenging and stressful for children, families, and teachers, which is evidenced in one teacher’s response: “*It is hard because we have one or two teachers constantly running out to receive children from the car or take them to the car*”.

Fifth, teachers reported that they experienced too many changes to their routines, staffing, and grouping and were never certain of what to expect the next day. Due to low enrollment and staff shortages, centers/schools had to lower teacher–child ratios and merge different groups (resulting in challenges teaching multiple age groups). Feelings of being overwhelmed and uncertain sometimes stemmed from a lack of clear communication and guidance at the program, district, state, and national levels.

In response to the challenges they experienced, the in-person teaching group reported that they needed: (a) more resources, supplies, staff, and testing to cope with COVID-19-related challenges; (b) more financial support for the center/school, including better or additional (hazard) pay and benefits for teachers; (c) clearer, more equitable, and more consistent regulations and communication; (d) more emotional support, such as appreciation, respect, and acknowledgement; and (e) more positive attitudes and hope for their situation.

The online teachers also experienced many challenges, but these were substantially different from the challenges reported from the in-person teachers. They identified the following as their major challenges: (a) difficulty supporting children’s learning via online teaching; (b) difficulty with parent involvement; (c) technology issues; (d) social isolation/feeling of disconnection; and (e) barriers to resources and preparation for online teaching.

Similar to the in-person teaching group, this group of teachers experienced challenges supporting children through online teaching. Many teachers in this group mentioned that it was difficult to get children to participate and engage in the lessons, and the overall rate of attendance and participation was low. Teachers also found it difficult to make activities engaging and developmentally appropriate through the online platform. These challenges often led to concerns about whether and how much children were learning through this format and how children were doing at home (e.g., they might miss signs of neglect or abuse). Some teachers commented that online teaching made it even more challenging to engage and support dual-language learners and children with special needs and expressed concern that online teaching would undermine equity and exacerbate learning gaps for children from marginalized groups. This was clearly evidenced in this teacher’s response:

*Connecting with young children over the screen is HARD. We are not able to address their needs/goals regarding behavior or social emotional play skills. It’s all artificial. We are missing out on teaching them and addressing their needs during critical windows of development. Many of these kids just started getting services (special needs) and are going to kindergarten in the fall. It’s just yucky all around*.

Second, difficulty with parent involvement was a common issue. Teachers acknowledged that the online teaching format relied heavily on parent involvement and empathized with parents that this added a significant burden for them. They often found it difficult to engage families already distressed from various hardships and additional work demands in their children’s learning at home. One teacher expressed this concern:

*Parents were totally overwhelmed by more demands and school expectations and they weren’t able to do all the Zoom meetings and lesson activities. I felt like I tried to focus most of my support on the parents and let them know that I believe they are doing their best and that is OK*.

Third, unlike in-person teaching, the availability of and access to technology was an inevitable concern for online teaching. The technology-related issues that teachers experienced included limited access to the internet, unstable internet connectivity—especially for children living in rural or impoverished areas—and access to an appropriate computer and electronic devices necessary for online learning. As one teacher put it, “*The biggest challenge for me is attendance because they do not have access to the internet. Because I teach in a low socio-economic area, many of my students do not have access to technology and/or the internet*”.

Fourth, many teachers found it difficult to engage children in social interactions and felt disconnected from their children in the online teaching format. They mentioned that it was difficult not to be physically present for children, show affection (e.g., they cannot hug, make direct eye contact, and play with children), and build and foster relationships. Lastly, teachers noted limited resources and a general lack of preparation necessary for quality online teaching. Although there were some online trainings offered to them, teachers still felt unprepared and ill-equipped to teach online. They felt that it was difficult to find the right resources, feel effective, and maintain accountability.

To address these challenges, online teachers requested the following for improvement: (a) better access and recourse for online teaching, including technology and internet accessibility; (b) more parent involvement and better ways of engaging them; (c) improved curriculum and format optimized for online learning; (d) clearer and more consistent guidelines and communication, and (e) more training on online teaching. Analysis of the quantitative data (see Table 4 below) revealed no differences among our measures of working conditions by teaching modality.

### 3.2. Early Childhood Teachers’ Well-Being by Teaching Modality during the Pandemic

Overall, a substantial number of teachers reported poor psychological and physical well-being. This is evidenced in both quantitative and qualitative responses. The content analysis identified the most frequently reported responses about perceived changes in well-being (see Table 3). Overall, early childhood teachers experienced remarkably similar psychological and physical well-being-related issues across the three teaching modalities. Across questions on psychological and physical well-being, five themes emerged: (a) more stress, anxiety, and fear of becoming ill with COVID-19; (b) weight gain and lack of physical activity; (c) increased feeling of social disconnection, depression, and sadness; (d) increased concerns about other illnesses as an existing or new health condition; (e) financial concerns.

First, stress, anxiety, and fear of becoming ill with COVID-19 were the most common across the three groups. Those who experienced negative changes in well-being reported high levels of anxiety and stress, often related to fear and uncertainty due to COVID-19. Anxiety and stress were replete and by far the most common challenge to well-being among our sample of teachers. Regardless of teaching modality, it was common that teachers were anxious and fearful of the possibility of contracting COVID-19. Stress was also frequently reported from teachers who were now juggling online teaching while also helping their own children with online learning. A teacher in the online group reported:

*There is constant anxiety. Not knowing if I will lose more families; if I will be able to replace the ones I did lose any time soon; if I or someone in my family gets the virus and unemployment errors cause delays, will we survive financially*?

Second, weight gain and lack of physical activity were also prevalent concerns among early childhood teachers during the pandemic. This was common for all three groups, but it was especially prevalent for online teachers. They reported weight gain resulting from stress eating, unhealthy food choices, being in close proximity to food all the time, and increases in sedentary behavior. As one teacher noted:

*I have gained about 20 pounds (during the pandemic). The added weight has affected my mobility. I have an increased level of discomfort with what formally was very mild aches and pains. I haven’t been to the doctor, but I’m sure my blood pressure is out of whack. These may be contributing to what I believe is mild depression*.

Teacher weight gain was also partially due to increased sedentary behavior and a lack of physical activity, and this was a concern across modalities. As one teacher in this online group remarked, “*During the school year, I feel as if I prioritized my diet and exercise more. By spending 7.5 h a day (or more online), my body is very sore after sitting all day*”.

Third, feelings of social disconnection, depression, and sadness were the next most prevalent theme across the groups. Teachers felt socially disconnected from their coworkers, students, friends, and family. Social disconnectedness was described as being lonely and feeling distant from loved ones (e.g., not being able to meet their family and friends). Missing their students was common for online teachers and teachers whose sites were closed. One teacher stated, “*I miss the children, I miss the job that I love, I miss my routine, and I miss interacting with my coworkers*”.

Depression and hopelessness were commonly reported well-being-related challenges, especially for the online and school closed groups. Teachers connected these feelings of sadness and depression to being out of their routine, missing their kids at school, and feeling trapped—a finding that was not shared by the in-person teachers. A teacher teaching online wrote:

*I’m a very productive person typically. Being cooped up at home has been very challenging to my happiness. Some days, I don’t want to get off the couch. Some days, I am very productive and get everything done! I try to spend time in my yard and growing things in my garden. But when I run out of things to do, I get sad… It is difficult for me to put my foot down and refuse to go out when pressured by others as well*.

Fourth, increased concerns about other illnesses as an existing or new health condition were often noted as a challenge in physical well-being during the pandemic. Illnesses incurred pre-pandemic were exacerbated, and new ones arose. One teacher reported, “*I have Type 1 diabetes, and the stress causes my blood sugars to run high*”. Another teacher says, “*stress has brought on migraines, which I had not previously experienced, and I’ve been having frequent chest pains*”. Other illnesses included insomnia, reduced energy, restless sleep, and poor sleep patterns.

Fifth, financial concerns were also identified and there were slight differences among the three teaching modalities. Financial concerns seemed to be a more common issue for the in-person group and the school closed group. One teacher puts it, “*It’s very frustrating financially. I know many people who are making about double in unemployment benefits than what we are making having to go to work every day and risk our health. My friend is getting over $1000 a week from unemployment, and we won’t be getting any raises this year due to COVID. Teachers deserve competitive pay!*” However, financial issues were not as prevalent for the online teaching group.

In general, the quantitative analysis corroborated the findings of the content analysis on the range of challenges that teachers experienced related to psychological and physical well-being described above. With respect to psychological well-being, our quantitative data revealed that 31% of teachers in the sample reported doctor-diagnosed anxiety and 23% reported doctor-diagnosed depressive symptoms (see Table 2), with 35% having depressive symptom scores reaching clinical levels (based upon a recommended cut-off score of 11 for the shorter CES-D instrument). Forty-eight percent of teachers experienced somewhat or mostly negative changes in psychological well-being during the early months of the pandemic.

Regarding physical well-being, 20% of teachers reported experiencing somewhat or mostly negative changes in their physical health, with substantial numbers of teachers reporting chronic conditions such as being overweight or obese (72%), having high blood pressure (28%), or asthma (22%; see Table 2). With regard to physical activity, 63.56% indicated that they performed an average of 1.37 h of moderate to vigorous physical activities every week. From an ergonomic perspective, 78% of the study participants reported having at least one area of work-related pain, and half (50.29%) indicated that pain interfered to some degree with their work.

Our regression analysis of the main effects of teacher race, center type, and modality, which is displayed in Table 4, reveal that ergonomic pain was highest for in-person teachers, as compared to teachers whose sites were closed, *β* = 0.155, *SD* = 0.076, *p* < 0.05. Regarding professional well-being, more than fifteen percent of teachers (15.40%) in the sample reported that they wanted to leave as a result of the current situation. Furthermore, work commitment was lower for in-person teachers, *β* = −0.298, *SD* = 0.075, *p* < 0.01, than teachers whose centers were closed, *β* = 0.203, *SD* = 0.081, *p* < 0.05, and there were similar differences among these two groups for intent to leave, with in-person teachers reporting higher intent to leave, *β* = 0.219, *SD* = 0.075, *p* < 0.01, than their closed counterparts, *β* = −0.128, *SD* = 0.081, *p* = n.s. Teachers teaching virtually showed no distinct differences with respect to professional well-being as compared to teachers whose schools were closed.

Among those who wanted to leave, the major reasons for thinking of leaving were health concerns and a fear of contracting the COVID-19 or passing it to others (41.50%). The second most frequent reason was dissatisfaction with their current teaching assignment (e.g., they did not want to teach online or in person as it is difficult to work with young children in a developmentally appropriate way under these circumstances, etc.) and/or additional job demands (16.98%). Further, teachers wanted to leave because they did not feel that their field had provided enough job security amidst the crisis and were interested in looking for other career opportunities (15.85%).

With respect to teacher psychological well-being, while there were no differences between teachers by modality, we did find racial differences among teachers and these effects held in our moderation analysis (see Table 5), which we discuss below. First, the pattern of difference in psychological well-being was pronounced for Black and Hispanic teachers whose centers were closed. On average, Black teachers had lower personal stress, *β* = −0.545, *p* < 0.01, depressive symptoms, *β* = −0.458, *p* < 0.01, and secondary trauma, *β* = −0.299, *p* < 0.05, as compared to White teachers whose centers were closed, *β* = 0.170, *p* < 0.05, *β* = 0.284, *p* < 0.01, and *β* = −0.085, n.s., respectively. The same was true for Hispanic teachers in comparison to White teachers: Hispanic teachers reported lower personal stress, *β* = −0.306, *p* < 0.01, depressive symptoms, *β* = −0.355, *p* < 0.01, and secondary trauma, *β* = −0.395, *p* < 0.01, and reported higher life satisfaction than White teachers whose centers were closed, *β* = 0.404, *p* < 0.01. White teachers whose centers were closed had the poorest psychological well-being of the different racial groups—in particular, they had the highest levels of personal stress, *β* = 0.170, *p* < 0.05, and depressive symptoms, *β* = 0.284, *p* < 0.01, and Black and White teachers shared similarly low levels of life satisfaction, *β* = −0.35, *p* < 0.05, for White, *β* = 0.003, n.s., for Black, respectively.

However, these patterns shift when we look at those teachers teaching in person and virtually. Those Black teachers teaching in person exhibited large differences in personal stress, *β* = 0.425, *p* < 0.05, and depressive symptoms, *β* = 0.362, *p* < 0.10, from Black teachers whose schools were closed while, for White teachers, being in person or closed did not make a difference: personal stress, *β* = 0.001, n.s., depressive symptoms, *β* = −0.087, n.s., secondary trauma, *β* = 0.104, n.s., and life satisfaction, *β* = 0.059, n.s. Hispanic teachers teaching in person were similar to Black teachers, showing significantly lower life satisfaction than those whose centers were closed, *β* = −0.381, *p* < 0.05, and brief resiliency for those teaching virtually (*β* = −0.405, *p* < 0.05). Other notable findings for psychological well-being were the fact that teachers with an associate’s degree or higher reported marginally higher life satisfaction, *β* = 0.239, *p* < 0.01, and lower depressive symptoms, *β* = −0.111, *p* < 0.10. Public school teachers, as compared to childcare center/pre-K teachers, had higher life satisfaction, *β* = 0.279, *p* < 0.01, but also marginally higher secondary trauma, *β* = 0.366, *p* < 0.01.

With respect to professional well-being, there were also some notable findings. First, White teachers teaching in person had significantly lower work commitment, *β = *−0.197, *p* < 0.05, and marginally higher intent to leave, *β* = 0.153, *p* < 0.10. There were also a few differences with respect to physical well-being. As mentioned above, Black and Hispanic teachers at closed centers had marginally lower ergonomic pain than White teachers, *β* = −0.238, *p* < 0.10, *β* = −0.229, *p* < 0.05, *β* = 0.059, n.s., respectively, but Hispanic teachers teaching in person experienced sharply higher ergonomic pain as compared to White teachers, *β* = 0.443, *p* < 0.05, *β* = 0.059, n.s., respectively.

Finally, with respect to job demands by modality or race (or the interaction between the two), there was only one difference: teachers teaching in person reported higher physical job demands across the board, *β* = 0.163, *p* < 0.10, with no differences among racial groups. Other differences, not surprisingly, broke down by education level and center type; teachers with higher education levels reported marginally lower physical job demands but higher skill discretion and decision authority, *β* = −0.116, *p* < 0.10, *β* = 0.249, *p* < 0.01, *β* = 0.201, *p* < 0.01, respectively. Family childcare home teachers had high decision authority and Head Start teachers lower as compared to childcare centers/Pre-K, *β* = 0.826, *p* < 0.01, *β* = −0.339, *p* < 0.01, respectively. Public school teachers had higher skill discretion than childcare center/Pre-K teachers, *β* = 0.390, *p* < 0.01.

### 3.3. Addressing Challenges: Teachers’ Reported Needs for Support

In concluding our analysis, teachers were also asked to rank the top three items needed to support their well-being out of a list of 22 possible choices. We combined some categories in arriving at 12 overall themes/responses. Among these, the five most frequently listed responses included: (a) higher wages (16.81%); (b) more resources for health and well-being (15.84%); (c) more coaching, mentoring, and professional development (including comprehensive safety training, 12.58%); (d) more daily breaks and paid leave (11.17%), and (e) more support for children with behavioral challenges and special needs (10.54%).

## 4. Discussion

Dramatic shifts in working conditions occurred for ECE teachers during the pandemic, as some remained teaching in person, while others taught online, and still others whose sites were closed were not teaching at all. Even before the pandemic, the ECE workforce was often characterized as a marginalized group because of their exposure to poor working conditions and their heightened risk of diminished well-being [1,4,8]. We sought to understand if and how these conditions changed during the early days of the pandemic and the role of teaching modality as well as teacher and center characteristics in any differences we found. Thus, this study examined the challenges, risks, and needs for early childhood teachers’ work and well-being primarily as a result of these shifts in their work during the COVID-19 pandemic in the U.S. We sought answers to the following three research questions: (a) How did ECE teachers’ working conditions and well-being differ by teaching modality (i.e., in-person, online, closed school)? (b) Were there teacher demographic and center-based differences in teacher well-being? and (c) Did teaching modality moderate the relationship between demographic and center-based differences and teacher well-being? The following discussion is organized along the lines of these three questions.

### 4.1. Differences in Working Conditions and Well-Being by Modality

Overall, while our findings are in line with those in recent COVID-19 studies mostly focused on K-12 teachers [8,9,18,19,27,28], they extend our understanding of the pandemic’s effects on work and well-being by highlighting the unique effects that it has had on the early childhood teacher workforce. This was one of the first studies, to the authors’ knowledge, to take a holistic view of ECE teacher work and well-being (i.e., psychological, physical, and professional well-being) using a large sample of teachers serving birth through Kindergarten from nearly all states in the U.S. Our mixed-method approach provided both the depth and breadth needed to fully explore this complex issue and the story we captured is a complicated one—both in terms of the nature of the challenges faced across the three teaching groups as well as the differences between racial groups we found.

Generally speaking, our sample of ECE teachers experienced challenges working with children both in person and online, but most of the challenges they experienced were different in type and intensity due to the unique circumstances of these different teaching formats. One clear distinction between these two modalities was the continued physical demands for in-person teachers and increased demands for new skills to teach virtually for online teachers. In addition to the increased physical job demands and constantly changing guidelines and regulations, in-person teachers had more concerns about financial restrictions due to low enrollment, the concomitant possibility of school closure, and fear of contracting COVID-19 at work. In-person teachers’ financial concerns may be related to the fact that they are more likely to work in settings where income sources are not stable or are highly tuition-dependent (e.g., family childcare homes, private childcare centers). This echoes previous findings that, while the pandemic compromised ECE teachers’ financial stability across all settings during the pandemic, it is possible that it disproportionately affected childcare teachers [8,51].

Conversely, as expected, online teachers had different concerns about technological issues, social isolation, as well as a lack of resources and support needed for online teaching. Early childhood teachers often use technology as a teaching tool to some extent (e.g., showing a video) [52], but the challenges in learning this new mode of teaching brought increased new skill demands as they faced challenges engaging children and families in an online format. The quantitative data also corroborate these findings in that in-person teachers perceived more physical demands while online teachers reported that their jobs required new skills and training (e.g., technology integration for online teaching). Due to these unique challenges and demands, the perceived needs of in-person versus online teachers were necessarily different.

These findings are consistent with previous studies on the challenges that teachers have experienced during the pandemic [11,17,28,53,54]. However, the current study adds to the extant literature by demonstrating that these challenges can greatly differ by teaching modality. In addition, we extended the findings of previous studies by investigating teachers’ needs directly. For example, the current study found that in-person teachers and online teachers needed vastly different types of resources in order to perform their teaching tasks (e.g., more cleaning supplies and financial support for in-person teaching vs. better technology, resources, curriculum, and training for online teaching). To provide appropriate resources and support, it is important to first assess what the challenges are for teachers, centers, and schools who are having potentially very different pandemic working experiences. Resources and support will need to be tailored to better meet the needs of teachers working under these very different circumstances.

While approximately half of the participating teachers perceived negative changes in psychological well-being, only approximately 20 percent of teachers reported negative changes in their physical health. This indicates that the COVID-19 pandemic appeared to more acutely impact psychological rather than physical aspects of well-being for most teachers. This was noteworthy given that COVID-19 is a major health-related issue. What is clear from our data, however, is that serious concerns remain about the overall health and physical well-being of ECE teachers. Among those teachers who experienced negative changes during the pandemic, teachers across all three teaching modalities shared anxiety and fear of COVID-19 and social isolation/disconnection as the greatest challenges related to their well-being. Concerns about weight gain and lack of physical activity as well as chronic health conditions were prevalent and common among those who reported a change during the pandemic among the three groups—chronic health conditions that, in some cases, were shared by the majority of our sample, including obesity, high blood pressure, and asthma, and that place them at higher risk for contracting and becoming very ill from COVID-19 [25,55]. This is remarkable, particularly for those teachers who continue to teach in person. These findings support and add to the limited extant evidence on teachers’ concerns about diminished physical well-being during the pandemic [22,25,56].

Teachers in all three groups also had other issues in common, such as feelings of social disconnection, depression, and sadness, financial concerns, and additional job demands, although there were slight differences across the three groups. For example, financial concerns, additional job demands, and lack of resources and support were more prevalent challenges for the in-person teaching group than the others. Loss of purpose was more unique to online teachers, and concern for students was more common for school closed teachers, which may be related to fact that they were not able to directly interact with the children—an aspect of the job that has traditionally been a strong attractor to the profession.

### 4.2. Differences in Well-Being by Teacher and Center Characteristics, Modality

We found many common issues and challenges related to psychological and physical well-being across the three teaching groups from the qualitative analysis, but a more nuanced picture emerged from the quantitative analysis. Upon further examination of the effects of well-being by modality, we found them, to a degree, to be moderated by teacher race—in particular, for psychological and professional well-being. Our findings show that White teachers’ psychological well-being was poorer on average and, across the measures of well-being, tended differ less by teaching modality. Black and Hispanic teachers, on the other hand, experienced large differences in psychological well-being indicators such as life satisfaction, stress, depressive symptoms, and secondary trauma depending on whether their centers were closed or they were teaching in person. When their centers were closed, Black and Hispanic teachers had better well-being, yet looked similar to White teachers when they were teaching in person. In contrast, White teachers experienced lower professional well-being outcomes (lower work commitment and higher intent to leave) when teaching in person versus closed, but Black in-person and virtual teachers experienced even larger negative effects on work commitment than White teachers. These findings corroborate and extend prior work [29,31,32] in that the COVID-19 crisis has had a disproportionate impact on vulnerable people, yet our findings reveal that this impact was unique and complex in its effects on minoritized groups. Because few studies have examined race differences in well-being during the pandemic, there is little precedent for understanding these findings. What seems clear is that more concerted effort should be made to examine the differential effects of race on the pandemic ECE teaching experience.

### 4.3. Limitations

This study has some limitations. First, although the data were collected from a large group of teachers from 46 states in the U.S., they may not be representative of the national sample as the sample was not randomly drawn. It was also a one-time, cross-sectional study. Teachers, early childhood programs, and schools experienced and continue to experience rapid changes in COVID-19 prevalence, regulations, and guidelines as the pandemic continues, and this means that teachers’ experiences may have changed drastically since our study was conducted. The present study only captured a snapshot of the early phase of the pandemic. The field would benefit from rigorous, longitudinal studies of the impacts of the pandemic on early childhood well-being to examine the “real impact of the COVID-19 pandemic and real changes”.

Second, we did not examine hybrid teaching as a separate modality; teachers in our study were asked to select one of the three categories. Because our data were collected in the early phase of the pandemic, the hybrid option was not as prevalent, but became more so in the later phases of the pandemic. Including this group would likely reveal yet another set of unique challenges, which would warrant further investigation. Third, we relied solely on teacher reports on their perceptions of work and well-being during the pandemic. Although it is valid to use self-reports in this case, adding an objective measure such as a direct assessment or a doctor’s report on health conditions would provide a more accurate picture of teachers’ experiences and well-being. Fourth, our intent in this study was to examine challenges in working conditions and well-being; however, in order to offer clearer implications for improving teacher well-being, future studies should include a larger set of center-level context and climate variables such as effective communication, leadership, and professional development support.

### 4.4. Implications for Practice and Policy

The COVID-19 pandemic has disrupted all sectors of the workforce, particularly those considered as the frontline workers. Our study provides ample evidence of the various challenges and high demands that early childhood teachers face that would have a negative impact on their well-being and work during the pandemic. This calls for additional resources and support to address the urgent needs. Stressed, overworked, and depressed teachers have little hope of meeting the needs of similarly stressed, traumatized, or otherwise needy young children. Support and resources to improve teachers’ psychological and physical well-being could include a physical wellness program, mental health services, and increased time for breaks and physical activity.

Regarding concerns about a high rate of obesity, weight gain, and lack of physical activity and energy, the American College of Sports Medicine [57] recommends that healthy adults aged 18–65 years should engage in moderate-intensity physical activity that increases the heart and respiratory rates for a minimum of 30 min five days per week. Given the evidence from recent studies, it is imperative for early childhood teachers to not only encourage physical activity in children but engage in it themselves throughout the day. This will be extraordinarily difficult for ECE teachers, who are typically not afforded time in the day to even take a 15 min restroom break. While additional funding will not solve these challenges alone, it could allow for the hiring of additional staff to help cover classrooms so that teachers may have time for breaks, exercise, or other leisure activities.

What is clear is that these supports cannot be monolithic; they need to be tailored to the unique challenges, demands, and needs at multiple levels and across various settings (e.g., centers, family childcare homes, public schools, online settings) and sectors of the early childhood workforce. This also includes increased training and support to prepare teachers for different teaching modalities. In particular, support needs to prioritize teachers teaching in person, who face more job demands and health risks and likely come from family childcare homes and private childcare centers that do not have stable funding sources. Finally, we need to more closely investigate the differential impacts of pandemic working conditions on the well-being of different racial groups, as this study revealed some evidence that these differences were consequential.

More important than the realization of the incredible stress and strain we have placed upon the teachers of our young children is the need to acknowledge the sacrifices for the greater good that these teachers have given and provide these much-needed supports as a gesture of appreciation for these incredible sacrifices. In doing this, we also need to better recognize these frontline workers as essential—those workers who in many cases have risked, and continue to risk, their well-being and health to support children and families during these difficult times. This shift in perception and recognition, coupled with program, policy, and funding changes, can help to prioritize the needs of schools and ECE teachers, which will support their work and well-being.

## 5. Conclusions

While we see signs that the pandemic is waning—vaccinations are increasing and COVID-19 variants have receded for the moment—inviting a return to “normal”, this does not absolve the field from ensuring that teachers are better prepared in the future to meet unique needs, whether pandemic-related or otherwise. For example, there has been an influx of online teaching in ECE due to the pandemic and these learning formats will likely persist long after the pandemic has subsided [7]. In our present circumstances, educators have learned how to adjust to weather this tumultuous time. It may not be a question as to if but when conditions worsen that they need to once again put these skills to action. For teachers working in person or online moving forward, we must continue to help them to address the physical and emotional demands of the profession. Even prior to the pandemic, we knew that teachers were suffering from a number of physical and health-related ailments due to stressful and strenuous working environments, poor wages and health benefits, and a lack of breaks [2,7]. Supporting teachers with plentiful and appropriate resources is an important pre-emptive step to ensuring that acute demands do not push our teachers and the profession over an edge from which there is little hope of recovery. The time is ripe to begin to seriously address the needs of this workforce to ensure that our educational systems are prepared for other challenges.

## Figures and Tables

**Table 2 ijerph-19-04919-t002:** Psychological and physical well-being of 1434 ECE teachers during pandemic.

Categories	Percentage OR Mean (Range)
Sample Demographics	
Teachers with Associates Degree or Higher	73%
Teachers by Center Type	
Family Childcare Home	5.6%
Childcare Center/Pre-K	34.4%
Head Start	42.8%
Public School	14%
Private School	3.2%
Teachers by Modality	
In Person	29.2%
Online	27.5%
Closed	43.3%
Teacher Race	
Black	14.3%
Hispanic	21.4%
White	58.3%
Other	6%
Psychological Well-Being	
Changes in Psych. Well-Being Due to Pandemic	
Mostly Positive	7.7%
Somewhat Positive	11.2%
No Change	32.8%
Somewhat Negative	43.0%
Mostly Negative	4.7%
Diagnosed with Depression	23%
Brief Resiliency	2.91 (1–5)
Life Satisfaction	24.64 (5–35)
Perceived Stress	25.60 (1–47)
Depressive Symptoms	8.38 (0–30)
Secondary Trauma	19.29 (3–50)
Physical Well-Being/Health	
Teachers with At Least One Area of Ergonomic Pain	79%
Total Reported Ergonomic Pain ª	3.49 (0–19)
In-Person Teachers	3.91 (0–15)
Online Teachers	3.59 (0–19)
Closed Teachers	3.08 (0–14)
Changes in Health Due to Pandemic	
Mostly Positive	5.2%
Somewhat Positive	7.7%
No Change	66.5%
Somewhat Negative	17.7%
Mostly Negative	1.9%
Currently Overweight or Obese	76%
Diagnosed with Anxiety	31%
Diagnosed with High Blood Pressure	26%
Diagnosed with Asthma	20%
Experiencing Food Insecurity	1.07(0–5)
Professional Well-Being	
Work Commitment	8.51 (1–10)
Intent to Leave	2.19 (1–5)
Job Demands and Resources	
Physical Job Demands	2.91 (1–5)
Skill Discretion	3.93 (1–5)
Decision Authority	3.35 (1–5)
Employer-Paid Health Insurance During Pandemic	59%
Wages—Full Pay During Pandemic (Partial Pay)	83% (12%)

Note. ª is an indicator of the product of both number of affected areas (max of 5 areas) and severity of pain (from 0 = no pain to 4 = unbearable pain), for a max range of 20.

**Table 3 ijerph-19-04919-t003:** Content analysis summary: challenges in work and well-being.

In-Person Teaching Group			Online Teaching Group			School Closed Group		
Challenges in Work (or Job Demands)								
Rank	Freq	Content	Rank	Freq	Content			
1	19.57%	Working with children and families during a pandemic	1	34.52%	Difficulty supporting children via online teaching			
2	18.40%	Financial hardship	2	24.30%	Difficulty with parent involvement			
3	16.63%	COVID-19 fear and uncertainty	3	13.55%	Technology issues			
4	14.48%	Additional job demand	4	11.51%	Social isolation/feeling of disconnection			
5	13.89%	Too frequent changes in regulations and circumstances	5	9.21%	Barriers to resources and preparation for online teaching			
Needs for Work								
Rank	Freq	Content	Rank	Freq	Content			
1	19.91%	More COVID-19 resources	1	20.34%	Improved access and resources			
2	14.13%	Financial support	2	16.16%	Better parent involvement			
3	13.06%	More consistency	3	12.26%	Improved curriculum			
4	10.92%	None	4	9.96%	Improved communication and guidelines			
5	6.42%	More support for teacher	5	7.94%	More training			
Challenge in Psychological Well-Being								
Rank	Freq	Content	Rank	Freq	Content	Rank	Freq	Content
1	52.64%	Anxiety and fear of COVID-19	1	32.98%	Anxiety and fear of COVID-19	1	50.48%	Anxiety and fear of COVID-19
2	9.97%	Social disconnection	2	23.11%	Social disconnection	2	21.40%	Social disconnection
3	9.66%	Financial concern	3	10.92%	Depression and sadness	3	11.18%	Depression and sadness
4	5.61%	Additional job demands	4	6.09%	Loss of purpose	4	7.99%	Financial concern
5	4.67%	Lack of support	5	5.67%	Additional job demands	5	2.88%	Concern for students
Challenges in Physical Well-Being								
Rank	Freq	Content	Rank	Freq	Content	Rank	Freq	Content
1	22.68%	Stress/anxiety	1	31.82%	Weight gain	1	27.52%	Stress/anxiety
2	19.59%	Other illness syndrome	2	20.78%	Stress/anxiety	2	23.85%	Other illness syndrome
3	18.56%	Weight gain	3	14.29%	Other illness syndrome	2	23.85%	Weight gain
4	13.40%	Energy change	4	13.64%	Lack of physical activity	4	8.26%	Lack of physical activity
5	8.25%	Lack of physical activity	5	5.84%	Sleep change	5	7.34%	Sleep change
5	8.25%	Sleep change	5	5.84%	Energy change			

**Table 4 ijerph-19-04919-t004:** Main effects regression of psychological, physical, and professional well-being by modality and teacher race.

	Brief	Life	Perceived	Depressive	Secondary	Work	Intent to	Ergonomic	Physic. Job	Skill	Decision
	Resiliency	Satisfaction	Stress	Symptoms	Trauma	Commit.	Leave	Pain	Demands	Discretion	Authority
Constant	0.113	−0.308 **	0.139	0.241 **	−0.075	0.203 *	−0.128	0.012	0.042	−0.287 **	0.021
	(0.081)	(0.080)	(0.080)	(0.080)	(0.080)	(0.081)	(0.081)	(0.082)	(0.082)	(0.081)	(0.078)
≥Associates degree	−0.050	0.227 **	−0.054	−0.112	0.138 *	−0.082	0.089	−0.023	−0.118	0.244 **	0.200 **
	(0.064)	(0.064)	(0.064)	(0.064)	(0.063)	(0.065)	(0.064)	(0.065)	(0.065)	(0.064)	(0.062)
Childcare/Pre-K	(reference)										
Family childcare	0.044	0.103	−0.099	−0.081	−0.214	0.083	−0.057	−0.019	0.103	0.105	0.850 **
	(0.126)	(0.125)	(0.125)	(0.125)	(0.123)	(0.126)	(0.126)	(0.127)	(0.127)	(0.125)	(0.121)
Head Start	0.096	0.096	−0.097	−0.168 *	0.047	0.094	−0.068	−0.024	−0.011	0.168*	−0.337 **
	(0.074)	(0.074)	(0.074)	(0.074)	(0.073)	(0.074)	(0.074)	(0.075)	(0.075)	(0.074)	(0.072)
Public school	0.078	0.277 **	0.043	−0.106	0.366 **	−0.029	−0.019	0.110	−0.158	0.408 **	−0.149
	(0.096)	(0.096)	(0.096)	(0.096)	(0.095)	(0.097)	(0.096)	(0.097)	(0.098)	(0.096)	(0.093)
Private school	−0.201	0.179	−0.086	−0.163	−0.137	−0.053	−0.139	−0.541 **	−0.250	0.047	0.312 *
	(0.162)	(0.161)	(0.161)	(0.161)	(0.160)	(0.164)	(0.163)	(0.164)	(0.165)	(0.162)	(0.157)
School closed	(reference)										
School in-person	0.019	0.004	0.084	−0.003	0.089	−0.298 **	0.219 **	0.155 *	0.139	−0.038	−0.017
	(0.075)	(0.074)	(0.074)	(0.074)	(0.074)	(0.075)	(0.075)	(0.076)	(0.076)	(0.075)	(0.072)
School online	−0.081	0.011	0.081	0.100	0.097	−0.107	−0.034	0.052	0.046	0.076	−0.041
	(0.067)	(0.067)	(0.067)	(0.067)	(0.067)	(0.068)	(0.067)	(0.068)	(0.068)	(0.067)	(0.065)
White teacher	(reference)										
Black teacher	−0.299 **	−0.009	−0.359 **	−0.275 **	−0.194 *	−0.201 *	0.218 **	−0.200 *	0.042	−0.018	−0.009
	(0.081)	(0.081)	(0.081)	(0.081)	(0.080)	(0.082)	(0.081)	(0.082)	(0.082)	(0.081)	(0.079)
Hispanic teacher	−0.343 **	0.326 **	−0.296 **	−0.309 **	−0.467 **	−0.136	0.020	−0.092	0.024	−0.166 *	−0.174 *
	(0.074)	(0.074)	(0.074)	(0.074)	(0.073)	(0.075)	(0.074)	(0.075)	(0.075)	(0.074)	(0.072)
Other race	0.063	−0.162	0.088	0.113	−0.114	−0.158	0.291 *	0.158	0.007	−0.120	−0.040
	(0.115)	(0.115)	(0.116)	(0.115)	(0.115)	(0.118)	(0.116)	(0.117)	(0.118)	(0.116)	(0.112)

Note. All outcome variables standardized. Standard errors in parentheses. ** *p* < 0.01, * *p* < 0.05.

**Table 5 ijerph-19-04919-t005:** Moderation of teacher race and psychological, physical, and professional well-being outcomes by modality.

	Brief Resiliency	Life Satisfaction	Perceived Personal Stress	Depressive Symptoms	Secondary Trauma	Work Commitment	Intent to Leave	Ergonomic Pain	Physical Job Demands	Skill Discretion	Decision Authority
Intercept	0.0905	−0.353 *	0.170 *	0.284 **	−0.085	0.170 *	−0.115	0.059	0.027	−0.297 **	0.019
	(0.085)	(0.085)	(0.085)	(0.085)	(0.084)	(0.086)	(0.085)	(0.086)	(0.092)	(0.085)	(0.083)
≥Associates deg.	−0.043	0.239 **	−0.049	−0.111 †	0.144 *	−0.088	0.094	−0.029	−0.116 †	0.249 **	0.201 **
	(0.064)	(0.063)	(0.064)	(0.063)	(0.063)	(0.064)	(0.064)	(0.065)	(0.063)	(0.064)	(0.062)
Childcare Center/Pre-K	(reference)										
Family CC home	0.064	0.076	−0.116	−0.088	−0.241 †	0.073	−0.055	−0.001	0.105	0.126	0.826 **
	(0.126)	(0.126)	(0.126)	(0.126)	(0.124)	(0.127)	(0.127)	(0.128)	(0.099)	(0.126)	(0.122)
Head Start	0.089	0.097	−0.092	−0.166 *	0.045	0.083	−0.060	−0.019	−0.013	0.166 *	−0.339 **
	(0.073)	(0.073)	(0.073)	(0.073)	(0.073)	(0.074)	(0.074)	(0.075)	(0.081)	(0.074)	(0.071)
Public school	0.054	0.279 **	0.029	−0.117	0.366 **	−0.020	−0.035	0.104	−0.157	0.390 **	−0.130
	(0.097)	(0.096)	(0.096)	(0.096)	(0.096)	(0.098)	(0.097)	(0.098)	(0.098)	(0.097)	(0.094)
Private school	−0.221	0.194	−0.103	−0.183	−0.140	−0.041	−0.148	−0.551 **	−0.246	0.035	0.321 *
	(0.162)	(0.161)	(0.162)	(0.162)	(0.161)	(0.164)	(0.164)	(0.164)	(0.174)	(0.163)	(0.157)
School closed	(reference)										
School in-person	0.017	0.059	0.001	−0.087	0.104	−0.197 *	0.153 †	0.074	0.163†	−0.058	0.015
	(0.090)	(0.089)	(0.090)	(0.089)	(0.089)	(0.090)	(0.090)	(0.091)	(0.093)	(0.090)	(0.087)
School online	0.011	0.076	0.065	0.056	0.108	−0.086	−0.009	−0.005	0.066	0.136	−0.087
	(0.088)	(0.088)	(0.088)	(0.088)	(0.088)	(0.089)	(0.089)	(0.090)	(0.089)	(0.089)	(0.086)
White teacher	(reference)										
Black	−0.335 **	0.003	−0.545 **	−0.458 **	−0.299 *	0.009	0.152	−0.238 †	0.056	−0.033	−0.084
	(0.122)	(0.121)	(0.122)	(0.121)	(0.121)	(0.123)	(0.122)	(0.124)	(0.138)	(0.122)	(0.118)
Hispanic	−0.251 *	0.404 **	−0.306 **	−0.355 **	−0.395 **	−0.107	0.004	−0.229 *	0.044	−0.149	−0.144
	(0.100)	(0.100)	(0.099)	(0.100)	(0.099)	(0.101)	(0.101)	(0.102)	(0.102)	(0.101)	(0.097)
Other	0.153	0.042	−0.004	0.055	−0.066	−0.081	0.213	0.153	0.121	−0.002	−0.012
	(0.176)	(0.178)	(0.178)	(0.178)	(0.177)	(0.180)	(0.178)	(0.181)	(0.184)	(0.179)	(0.173)
Black * In-person	0.073	0.117	0.425 *	0.362 †	0.264	−0.357 †	0.187	0.042	−0.019	0.04	0.115
	(0.187)	(0.187)	(0.188)	(0.187)	(0.185)	(0.189)	(0.187)	(0.189)	(0.197)	(0.188)	(0.182)
Hispanic * In-person	0.143	−0.381 *	0.116	0.242	−0.228	−0.286	0.134	0.443 *	−0.046	0.211	−0.245
	(0.187)	(0.187)	(0.187)	(0.187)	(0.186)	(0.189)	(0.189)	(0.191)	(0.201)	(0.188)	(0.182)
Other * In-person	−0.256	−0.186	0.131	−0.036	−0.194	−0.229	0.332	0.001	−0.193	−0.174	−0.180
	(0.263)	(0.263)	(0.263)	(0.263)	(0.262)	(0.267)	(0.265)	(0.268)	(0.259)	(0.265)	(0.257)
Black * Online	0.059	−0.189	0.204	0.268	0.099	−0.384 †	0.012	0.077	−0.021	0.003	0.173
	(0.204)	(0.203)	(0.205)	(0.202)	(0.200)	(0.204)	(0.203)	(0.205)	(0.227)	(0.203)	(0.196)
Hispanic * Online	−0.405 *	−0.039	−0.099	−0.034	−0.123	0.138	−0.079	0.187	−0.027	−0.197	0.061
	(0.159)	(0.159)	(0.159)	(0.159)	(0.158)	(0.161)	(0.160)	(0.162)	(0.154)	(0.161)	(0.156)
Other * Online	0.002	−0.596 *	0.202	0.299	0.103	0.023	−0.158	0.001	−0.198	−0.250	0.159
	(0.298)	(0.298)	(0.298)	(0.298)	(0.302)	(0.308)	(0.296)	(0.303)	(0.263)	(0.300)	(0.291)

Note. All outcome variables standardized. Standard errors in parentheses. ** *p* < 0.01, * *p* < 0.05, † *p* < 0.10.

## Data Availability

The data presented in this study are available on request from the corresponding author. The data are not publicly available due to privacy/confidentiality issues.

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
