# Peer review of "Challenges in Working Conditions and Well-Being of Early Childhood Teachers by Teaching Modality during the COVID-19 Pandemic"

_ijerph, 2022, doi:10.3390/ijerph19084919_

Round 1

Author Response

Please see the attached response letter. 

Reviewer 2 Report

Review of manuscript: ijerph-1653749

Recommendation:        accepted

Comments to Authors

The manuscript “Challenges in Working Conditions and Well-Being of Early Childhood Teachers by Teaching Modality During the COVID-19 Pandemic” by Kyong-Ah Kwon, Timothy G. Ford, Jessica Tsotsoros, Ken Randall and Adrien Malek-Lasater, evaluates the effect of COVID-19 pandemic on early care and education teachers' working conditions and physical, psychological, and professional well-being using a national sample of 1,434 ECE teachers in the U.S. The manuscript is related to an interesting topic. The study is very complex, on an extensive data set, statistically processed relatively simple and clear. The conclusions are interesting and useful, in order to analyze the effects of a crisis situation on teaching efficiency and can be a starting point in addressing new situations and implementing measures to streamline the education process. The paper is well written and organized, with a deep discussion of the results to give significance of the results found. Therefore, I recommend its publication in International Journal of Environmental Research and Public Health.

I have only a few minor remarks:

Table 1: row 4, Life satisfaction, column 4, …. 1(strongly disagree) to 7 (strongly disagree)… please correct

Table 1: row 5, Obesity, column 4, please correct BMI<18.5 and BMI>=30

Table 4: please explain the abbreviation for OLS

Author Response

(The authors gave the same response as above.)

Reviewer 3 Report

Revise carefully all references, as there are sometimes data missing (e.g. date in 116).

Be sure to define all acronyms before using them in the text (e.g. SES) as non-specialized readers may not be familiar with all of them. Even well-known acronyms, such as GED should always be spelled out the first time they are used in the text.

Results are complete and interesting. Nevertheless, try to facilitate their reading and provide forms to present them in a summarized and clear way.

Revise Table 1 concerning formatting (e.g. use of alpha in words or as a symbol, font size…). Try to make a short and homogeneous description of the column about “Instrument characteristics”, as some redundant information is observed. A homogeneous presentation, that addresses the same aspects and organized in an easy way (e.g. bullets) for all instruments would facilitate the reading. Similarly, try to facilitate the reading of data in Table 2, where different numbers mean different things and try to give a sense of the actual value of them (e.g. is 8.38 high or not?). Would it be possible to present them in percentages or normalized to 10 or any other number?

Try to avoid ambiguous sentences, such as “After cleaning the data” (246), explaining what you actually did or, if it is not relevant, omitting them. Similarly, support affirmations with references or data (e.g. “Even before the pandemic, the ECE workforce was often characterized as a marginalized group because of their exposure to poor working conditions and their heightened risk of diminished well-being.”  - 583)

A conclusions section is missing. Try to directly address the research questions.

Author Response

(The authors gave the same response as above.)

Reviewer 4 Report

Dear Authors

Hope to find you well.

Your research is very pertinent and I begin to congratulate you for that, teacher of several cycles have been exposed to unimaginable challenges but those who work with children were affected with problems of inconsistent, contradictory and unclear guidance and rules in the early childhood education arena. Although this issue is not new in research, it is always welcome when it brings more results that allows to deal with similar situations.

Probably many of these challenges will end when the pandemic is over, but the effects remain and we don't know if we will be exposed to similar situations in the future, so I agree that we have to think about political support strategies, both in terms of conditions and preparation for different educational modalities.

I think the article is good but there are some small things that I suggest to improve, for example:

line 16-17 - This phrase must be reviewed, its not well constructed, so it "Given the unique challenges and risks teachers may have faced by teaching modality (i.e., in-person, online, closed school), we also explored these differences" maybe it should be "We also explore the differences between in-person, online and closed school, given the unique challenges and risks teachers may have faced by teaching with different approaches".

You should revised the text and improve the writing to make it more clearer, some sentences are a bit long and your comprehension and objectivity lose with it, although this is not a reason for loss of quality 

Also you should have a  conclusion section to answer the initial questions of this study.

Wish you all the best

Author Response

(The authors gave the same response as above.)
